# Molecular Evolution of Tubulins in Diatoms

**DOI:** 10.3390/ijms23020618

**Published:** 2022-01-06

**Authors:** Kirill V. Khabudaev, Darya P. Petrova, Yekaterina D. Bedoshvili, Yelena V. Likhoshway, Mikhail A. Grachev

**Affiliations:** Limnological Institute, Siberian Branch, Russian Academy of Sciences, 664033 Irkutsk, Russia; onexkirill@gmail.com (K.V.K.); daryapetr@gmail.com (D.P.P.); likhoshway@mail.ru (Y.V.L.); grachev@lin.irk.ru (M.A.G.)

**Keywords:** microtubules, α-, β-, and γ-tubulins, phylogeny, morphology and evolution of diatoms

## Abstract

Microtubules are formed by α- and β-tubulin heterodimers nucleated with γ-tubulin. Tubulins are conserved eukaryotic proteins. Previously, it was shown that microtubules are involved in diatom silica frustule morphogenesis. Diatom frustules are varied, and their morphology is species-specific. Despite the attractiveness of the problem of elucidating the molecular mechanisms of genetically programmed morphogenesis, the structure and evolution of diatom tubulins have not been studied previously. Based on available genomic and transcriptome data, we analyzed the phylogeny of the predicted amino acid sequences of diatom α-, β- and γ-tubulins and identified five groups for α-tubulins, six for β-tubulins and four for γ-tubulins. We identified characteristic amino acids of each of these groups and also analyzed possible posttranslational modification sites of diatom tubulins. According to our results, we assumed what changes occurred in the diatom tubulin structures during their evolution. We also identified which tubulin groups are inherent in large diatom taxa. The similarity between the evolution of diatom tubulins and the evolution of diatoms suggests that molecular changes in α-, β- and γ-tubulins could be one of the factors in the formation of a high morphological diversity of diatoms.

## 1. Introduction

Microtubules (MTs) are major components of the eukaryotic cytoskeleton. Although MTs are involved in a variety of diverse processes ranging from intracellular transport to morphogenesis, their protein building blocks, called tubulins, are among the most well-conserved proteins [1,2]. Tubulin is a subject of study in various fields of molecular science, from cell biology [3] to evolution [4]. Knowledge of the structure and function of tubulins has practical applications in the development of new drugs for medicine [5,6] and agriculture [7]. 

The tubulin superfamily includes α-, β-, γ-, δ-, ε-, ζ-, and η-tubulins [4]; the evolution of this superfamily has been shaped by intense recent duplications [8]. The high degree of similarity between the bacterial filamentous temperature-sensitive protein Z (FtsZ) and the eukaryotic family of tubulins [9,10] has led to the inclusion of FtsZ (along with other prokaryotic tubulin-like proteins [11]) in the tubulin superfamily [12,13], and FtsZ is thought to be a tubulin ancestor [14]. The α-, β- and γ-tubulins are present in all eukaryotic phyla, whereas δ-, ε-, ζ- and η-tubulins are restricted to animals, fungi and protists [15,16,17] and have been lost in some phyla [8]. Phylogenetic analysis has shown that plant α- and β-tubulins can be separated into distinct subclasses [1,18]. In animals, the α- and β-tubulin genes have undergone duplication, encoding separate isoforms [19,20]. The various isoforms of α-, β- and γ-tubulins differ by their variable sites [21,22] mostly at the C-terminal ends, and the different isoform structures affect MT dynamics [23,24]. In addition to these duplications, the diversity of tubulin isoforms can be further increased by the application of post-translational modifications [11,22]. 

α- and β-tubulin monomers form a dimeric protein that serves as a basic building block for microtubules [13,15], and γ-tubulin is necessary to initiate their assembly [15,25,26]. Although the three-dimensional structure of the tubulin dimer has been studied [27,28,29,30,31,32], the roles played by some amino acid residues in both the α- and β-tubulin chains in model organisms are known [29].

For example, it is known that α- and β-tubulins consist of 10 β-chains (S1–S10) and 12 α-helices (H1–H12) connected by loops. These structures formed by three major domains: an N-terminal nucleotide-binding domain (1–205 amino acids [a.a.]), an intermediate domain (206–381 a.a.) and a C-terminal ligand-binding domain (from 382 a.a.; Figure 1) [24,33,34,35]. γ-tubulins have similar structure according to [36,37]. The C-terminal domain is also known as the C-terminal tail (CTT) [18], and its modifications affect protein diffusion along MTs [38] and alter dynein binding [1]. Amino acid substitutions in CTT may result in changes in protein function since this domain is responsible for interactions between microtubules and various intracellular components [1,39]. Tubulin heterodimers assemble longitudinally to form protofilaments. The H3-helice and M-loop (ML-surface) are involved in lateral interactions between heterodimers of neighboring protofilaments [40,41]. 

Diatoms are a unicellular eukaryotic group within the kingdom Chromista [42] that display a high diversity of morphologically distinct species [43]. The phylum Bacillariophyta consists of classes Coscinodiscophyceae (centric species with radial symmetry), Mediophyceae (centric species with radial and bipolar symmetry), and the bilaterally symmetric class Bacillariophyceae. Class Bacillariophyceae, in turn, consists of the subclass Bacillariophycidae (raphid pennates) and the araphid subclasses Fragilariophycidae and Urneidophycidae. Phylogenetic analysis of 4 markers expressed by araphid diatoms can be used to separate them into two major clades: (1) a small basal araphid clade, which includes families from Urneidophycidae and the genera *Neofragilaria* and *Asteroplanus*; and (2) a larger Fragilariophycidae clade, which includes other araphid pennates. This latter clade is thought to be a sister of the raphid pennates [44,45]. According to the timescale for diatom evolution based on four molecular markers [44,46], class Coscinodiscophyceae diverged from the remaining diatoms belonging to the subdivision Bacillariophytina 230 Ma (max). Within the Bacillariophytina the bi(multi)- polar centric class Mediophyceae diverged from the class Bacillariophyceae 218 Ma. The Bacillariophyceae radiated 190 Ma.

Numerous studies have attempted to determine the genetic and cellular mechanisms that underlie the differences in symmetry and the fine structures of diatom cell walls. Earlier studies have shown that this group of organisms expresses α-, β-, and γ-tubulins [8,47,48,49]. Exposure to microtubule inhibitors causes a variety of structural anomalies in diatom valves [50,51,52,53,54,55,56,57,58,59]. In other studies, tubulin has been shown to be involved during morphogenesis, and microtubules appear to form underlying layers during the formation of valves [60,61] and other specialized structures [56,57]. The role of microtubules is likely not limited to defining formations of structure; they are also likely to be involved in intracellular vesicular transport [50,62,63]. A hypothesis was put forward about the specific localization of aquaporins in silicalemma by MTs during the morphogenesis of silica structures [64].

Currently, the molecular mechanisms regulating of microtubules in diatoms have been insufficiently studied. For further experimental studies, it is necessary to analyze the primary structure of diatom tubulins and their regulating proteins. 

In this work, we performed a comparative and phylogenetic analysis of diatom α-, β- and γ-tubulins, including the description of their groups, duplications, and amino acid residues to define characteristic positions. Based on these data, we propose a model of diatom tubulin evolution.

## 2. Results

### 2.1. Tubulin Identification

Analysis of genomic data of diatoms *Fragilaria radians* (Kützing) D.M. Williams & Round, *Phaeodactylum tricornutum* Bohlin, *Thalassiosira pseudonana* Hasle & Heimdal, *Pseudo-nitzschia multiseries* (Hasle) Hasle, *Fragilariopsis cylindrus* (Grunow ex Cleve) Helmcke and Krieger showed the presence of α-, β- and γ-tubulin; in the genome of *Pseudo-nitzschia multistrata*, only the genes of α- and γ-tubulin were found. The genome of *P. multiseries* contained two α-tubulin genes with 95.58% sequence identity. *F. cylindrus* and *T. pseudonana* encoded two and three β-tubulin genes, respectively, with intra-genome sequence identities of 90.83% and 78.33 to 97.18%, respectively (Appendix A). 

The analysis of the MMETSP transcriptomic data resulted in the detection of all three tubulin subfamilies in the analyzed species (Appendix A). Of the known transcriptomes, 53 species encoded at least one α-tubulin transcript (17 encoded two, and 3 encoded three α-tubulin sequences, with varying intra-genome sequence identities) β-tubulins were identified in 54 species, with 14 species had 2 and 1 species had 3 sequences (which also featured various intra-genome sequence identities). γ-tubulin sequences were identified in 17 species, with no species encoding more than one transcript.

Mapping the structural conservation of diatom α- and β- tubulins (Figure 1; Appendix A) revealed that they are highly conserved throughout most of the sequence. In contrast, γ-tubulins are only conserved within the helices, whereas the loops featured multiple variable sites (Figure 1; Appendix A).

The length of the predicted a.a. sequences of diatom α-tubulins included in our data set varies from 445 to 468 a.a. Analysis of the alignment of all sequences including *Homo sapiens* and *Arabidopsis thaliana* (Appendix A) revealed that more than 50% of total a.a. number are conserved. Fourteen a.a. are conserved among diatoms, but differ from the same positions in out-group sequences. That is, these positions are conserved for diatoms and differ from the outgroup (see below “characteristic of diatoms”). Moreover, the positions are probably significant for the diatoms as the amino acids have retained conservatism. The amino acid residues are C75, Y87, I92, T141, S168, S172, T201, L209, V212, A232, T303, V378, L388 and A400 (Appendix A; Figure 2). 

Most of the differences in the diatom α-tubulin amino acid sequences were concentrated in the N-terminal domain and the CTT, this latter being enriched in negatively charged amino acids aspartate (D) and glutamate (E). Single amino acid polymorphisms could be detected along all of the sequence (Appendix A). 

The length of predicted a.a. sequences of β-tubulins of diatoms included into our data set varies from 422 to 466 a.a. Analysis of the alignment of all sequences considered, including *H. sapiens* and *A. thaliana* (Appendix A), revealed that more than 40% of a.a. positions are conserved. Differences among the β-tubulin predicted amino acid sequences of diatoms were typically found in variable positions along the entire sequence. Twelve positions are conserved for diatoms and differ from out group (see below “characteristics of diatoms”): M139, Y167, Q198, A201, A230, A232, T236, C248, C313, S323, A351, and W378 (Appendix A; Figure 3).

The length of the predicted a.a. sequences of γ-tubulins of diatoms included into our data set varies widely from 453 to 579 a.a. (Appendix A). The longest sequence was found in *P. delicatissima* (579 a.a.). This increase is caused by two insertions of varying length involved in connection with dimers of αβ-tubulins, located between T1 and T2 loops in the nucleotide binding domain (up to 60 a.a.), as well as in the region between S7 and H9 (up to 56 a.a.). Analysis of the alignment of all sequences considered including *H. sapiens* and *A. thaliana* revealed that 19 a.a. are conserved in diatoms and differ from those in organisms of the out-group (Appendix A; Figure 4).

### 2.2. Phylogeny 

Five groups were identified on the phylogenetic tree of diatom α-tubulins (Figure 2). The α1 and α3 groups are non-monophyletic, while the α2, α4, and α5 groups are monophyletic. Three of these groups (α1–α3) included centric species with radial symmetry. Group α1 consisted of species from the classes Mediophyceae and Coscinodiscophyceae. Group α2 was a small, heterogeneous group containing *Ditylum brightwellii*, *Minutocellus polymorphus*, *Extubocellulus spinifer* (class Mediophyceae), and *Aulacoseira subarctica* (class Coscinodiscophyceae). Group α3 (α3.1 and α3.2) is primarily formed by centric Mediophyceae, with two exceptions: *Dactyliosolen fragilissimus* (class Coscinodiscophyceae) and *Asterionellopsis glacialis* (class Bacillariophyceae, subclass Urneidophycidae). The other two groups consisted of pennate diatoms. Group α4 included araphid pennates from the subclasses Fragilariophycidae and Urneidophycidae (class Bacillariophyceae). Group α5 is found entirely among Bacillariophyceae taxa.

In some transcriptomes, the detected α-tubulin copies belonged to the same group, whereas some transcriptomes featured copies that were classified into multiple groups (Figure 2). For example, all of the α-tubulin sequences identified in *Amphiprora paludosa* (ApTa1, ApTa2, and ApTa3) belonged to group α5, whereas the α-tubulin sequences in *Ditylum brightwellii*, DbTa1a and DbTa2, were classified into α2 and α3 groups, respectively, and the sequence DbTa1 had an intermediate position (α1). Two sequences identified in *Triceratium dubium* (TdTa2a and TdTa2b) were placed in the out group, whereas a third sequence (TdTa1) also occupied an intermediate position (α1). 

The phylogen etic tree of diatom β-tubulins (Figure 3) consisted of six groups. β2 and β4 groups arise as a result of polytomy. Centric diatoms from the classes Mediophyceae and Coscinodiscophyceae, with multiradial and bipolar symmetry, formed the β1 group. The non-monophyletic β2 group consisted of centric diatoms from the classes Mediophyceae and Coscinodiscophyceae. Group β3 is formed by centric diatoms of class Mediophyceae. Group β4 is non-monophyletic and included centric diatom sequences from the family Chaetoceraceae (class Mediophyceae), a single araphid pennate diatom *Staurosira* complex sp. (subclass Fragilariophycidae, class Bacillariophyceae), a single centric diatom *D. fragillisimus* (class Coscinodiscophyceae) and single pennate *A. glacialis* (class Bacillariophyceae, Urneidophycidae). Group β5 represented by araphid pennate diatoms (subclass Fragilariophycidae, class Bacillariophyceae). Group β6 consisted of species subclass Bacillariophycidae, class Bacillariophyceae (pennate raphid diatoms), order Bacillariales, and pennate raphid *Phaeodactilum tricornutum* (Figure 3).

The γ-tubulins were separated into four groups (γ1–γ4) (Figure 4). The γ2–γ4 groups were also formed as a result of polytomy. The γ1 group was formed by two centric diatom species from the class Mediophyceae. The γ2 group included the centric diatoms from the family Chaetocerotaceae (class Mediophyceae) and a single Coscinodiscophyceae species, *Dactyliosolen fragillisimus*. The γ3 group contained centric diatoms from the class Mediophyceae, and pennate diatoms (class Bacillariophyceae) formed the γ4 group.

### 2.3. Analysis of Specific Amino Acids in Different Groups

As a result of sequence analysis within the phylogenetic groups, positions can be identified that are conserved within individual groups but differ from those of the outgroup (Appendix A). Some of the identified positions overlap between groups (Figure 2; Appendix A), while others are conserved within one specific group. The characteristic conserved positions of group α-tubulins are presented in Appendix A and Figure 2, so for the α1 group, 4 a.a. are characteristic (G19, S80, R370 and Q372) in N-terminal nucleotide-binding domain and intermediate domain. In the α2 group, conserved amino acids are located only in the N-terminal nucleotide-binding domain (Q19, S50), and in α4 in the intermediate domain (K372, M379). There are no such characteristic conserved amino acids in the α3 and α5 group.

Among the phylogenetically distinguished groups of β-tubulins, 11 positions are retained for each of them (Appendix A; Figure 3; Appendix A). The number of such positions is less than for α- and γ-tubulins. The largest number of conserved positions was found in β1 group (L84, M130, A145, V163, I170, C234, V315, T365, T371 and T377), while in other groups they were single and localized only in one of the domains N-terminal nucleotide-binding domain or the intermediate domain. It is interesting that for β5 group, the characteristic amino acids were found in the C-terminal domain. No characteristic conserved positions were found in group β4.

Due to the small number of diatom γ-tubulin sequences available for analysis it is quite conditional to distinguish conserved positions for γ-tubulin all groups (Appendix A; Figure 4; Appendix A). We did not define any specific conservative amino acids for γ1 group, since it consists of only two sequences (Appendix A; Figure 4). 

### 2.4. Analysis of Posttranslational Modification Sites

We performed a search for known post-translational modification sites in predicted amino acid sequences from five α-tubulin groups, intending to use these results in consequent theoretical and experimental reconstructions of cytoskeleton regulation on various stages of the diatom cell cycle.

It was shown that lysine in homolog positions to *H. sapiens* α-tubulin methylation and acetylation site is only retained in group α1. In other groups (α2, α3 and α5), some of the sequences were replaced by nonpolar or uncharged polar amino acid residues (Appendix A). The polyglutamylation sites of α-tubulin E443 and E445 [65] are retained in 35 and 46 cases of 83, respectively (Appendix A). Polyglutamylation site E443 in the similar positions is conserved in 35 out of 83 sequences [65]. The α1 group has the largest number of sequences with conserved polyglutamylation sites, and the fewest cases in the α4 and α5 groups in pennate species. The polyglycylation sites E446 and E448 in the sequences of diatom α-tubulins are almost not preserved. Glutamic acid is present in sequences 13 and 17, respectively, and it was not possible to isolate a group for which its presence would be necessary. In most cases, these positions contain aspartate, which also belongs to monoaminodicarboxylic (35 and 46 sequences, respectively) (Appendix A). The polyglycylation site of β-tubulin E437 is fully represented only in the β1 and β4 groups and in most of the sequences of the β2 group, while in the β5 and β6 groups it is replaced by aspartic acid in most of the sequences. In the positions corresponding to the polyglycylation site E439, in diatoms in most cases (44%) there is aspartate; glutamate is present less frequently (36.7%). The polyglutamylation site of E441 in β-tubulins is conserved only in the β3 group (retained in 13 sequences out of 14), while in other groups it is present only in some sequences. Sites of detyrosination in C-terminal tyrosine (Y) residue and polyglutamylation (addition of E to γ-carboxy group of E side chains and chain elongation by further addition of E residues) or polyglutamylation (addition of G to γ-carboxy group of E side chains and chain elongation by further addition of G residues) are conserved for all groups (Appendix A). However, glutamate (E) was replaced with aspartate (D) in several sequences (Appendix A). Sites that are homologous to *H. sapiens* phosphorylation and ubiquitinylation tyrosine Y432 and lysine K304, respectively, are also conserved for all groups (Appendix A). Palmitoylation (addition of long-chain fatty acid palmitate) site that is homologous to H. sapiens C376 is conserved in all groups of α-tubulin. Interacting amino acid pair responsible for filament rigidity (homologous to K60 and H283 for *H. sapiens*) is strictly conserved in α1, while other groups (α2-5) have some substitutions (Appendix A).

β-tubulin amino acid sequences show remarkably high identity, both within groups (β1-β6) and in general. A slightly higher variance is present in the CTT end (Appendix A). Analysis of functionally important modification sites shows that acetylation, polyamilation and phosphorylation sites homologous to *H. sapiens* K252, Q15, and S172 are strictly conserved. Polyglycylation site E435 is preserved in all groups except for substitutions to aspartate in several sequences. Polyglycylation site E438 remains conserved only in the β4 group; in the β5 and β6 groups, with a single exception, it is replaced by aspartate, in other groups it is non-conserved. In β4 glutamic acid (E) the former position is more conserved (again, with substitutions to glutamate); polyglycylation site E438 for β3 group is more variable and may contain monoaminocarbonic acids (G and A), an amide of a monocarbonic acid (Q), as well as aromatic amino acids (Y and T) (Appendix A).

Phosphorylation is the only type of modification described for γ-tubulin (S80, S131, T288 and S361). Analysis of the predicted diatom amino acid sequences shows that S131 and T288 are retained in all members of this group. Serine in position S361 is substituted with monoaminomonocarbonic acid alanine (A) for all diatom sequences. S80 is mostly replaced with glycine, another monoaminomonocarbonic acid, but it is not strictly conserved (Appendix A).

## 3. Discussion

### 3.1. Features of Diatom Tubulin a.a. Sequences

Tubulins are highly conserved, and the substitution of a single amino acid in these protein sequences can cause microtubule dysfunction and phenotypic changes [20,66]. Various tubulin isoforms are known to exist in nature [11,67], some of which have undergone subfunctionalization [1,68]. However, mutagenesis represents an essential tool in the search for novel approaches to the treatment of microtubule-related diseases [23,69]. 

Diatom genomes and transcriptomes have been found to encode α-, β- and γ-tubulins (Appendix A), as described by previous studies [8,47,48,49]. Only a few species appear to express γ-tubulin, although it is present in all genomes studied, which is likely due to the relatively low expression level of γ-tubulin compared with the α- and β-tubulins. γ-tubulin is a vital part of the acentriolar microtubule organization centre (MTOC) present in diatoms [47]. As this structure is only duplicated at certain stages of the cell cycle, γ-tubulin may not be expressed at other times, and thus not present in transcriptomic datasets. To collect more diatom γ-tubulin sequences, it is necessary to produce either genomic sequences or transcriptomes from synchronized cultures during MTOC duplication. It is likely that such datasets for all diatom species will include γ-tubulins. Long insertions in the T1-T2 nucleotide binding domain and the beginning of the second domain may be characteristic of diatom γ-tubulins (S5_Alignment3_Gamma_diatoms.fas). It is possible, however, that these insertions are removed during protein maturation.

Most amino acid substitutions identified within the diatom α- and β-tubulin sequences were found in the N-terminal domain and the CTT. The CTT is known to be positioned outside of the microtubule core and serves as a binding site for some microtubule-associated proteins (MAPs) [70,71]. The CTT has also been shown to undergo various modifications [39]. In addition, the CTT domain affects microtubule polymerization and depolymerization kinetics [33]. We suppose that the substitutions identified within this domain may affect some of these properties and the overall function of the microtubule apparatus. Insertions in this region have previously been shown to cause the repositioning of essential amino acids involved in GTP binding [72]. 

Tubulins are subject to numerous posttranslational modifications such as phosphorylation [73], acetylation [12,74], methylation [75], palmitoylation [76,77], ubiquitylation [78] polyamination [79], and detyrosination/tyrosination [80]. We have considered it crucial to study whether the sites of these modifications are preserved in diatom tubulin sequences. It was noted above that K40 in *H. sapiens* α-tubulin was an important modification site. Acetylation in this position protects microtubules from aging [81]. According to our results, this site is only preserved in the diatom group α1, and replaced with incompatible amino acids in other groups. Moreover, our findings revealed that in 93% of sequences, this site is preserved in the diatom group α1 and is absent in the group α4, which may affect the stability of the microtubules. It is likely that acetylation at this position is most significant for the α1 group. This suggests that, at least outside α1, microtubule longevity is either much lower than in humans, or regulated via a different mechanism.

Unlike K40, the C-terminal tyrosine and detyrosination site [80] and the glutamination and polyglutamination sites [82] are essential for microtubule flexibility and various microtubule-regulating signals. These sites are conserved for all diatoms (assuming that E-to-D substitutions are synonymous), pointing to an important role to microtubule functioning. Nevertheless, despite the synonymy of these amino acids, their substitution in some cases is still significant for the regulation of the properties of microtubules through glutamination [83]. It is possible that tyrosine phosphorylation site (Y432) [71] and lysine ubiquitination site (K304) [78] also retain their role. The site palmitoylation [76], a target of growth factor in human cells, is retained in all diatom α-tubulin groups. Low identity of this position in other groups suggests that growth factor is not an important microtubule regulator in diatoms. On the other hand, rigidity regulation mechanism involving K60 and H283 [81] is possible in diatoms, since these amino acids are conserved.

Most of β-tubulin modification sites are conserved (Appendix A), suggesting that diatom β-tubulin regulation is similar with other organisms [74,79,84]. The only exception is provided by polyglycylation sites (E435 [85] and E438 [86]) which are more variable (Appendix A). 

Only two phosphorylation sites are in diatom γ-tubulins, namely S131 and T288, but four phosphorylation sites (S80, S131, T288, and S361 [87]) are known among other organisms. Positions to be homologous to S80 and S361 contain entirely different amino acids (aminomonocarbonic alanine and glycine instead of oxymonoaminocarbonic serine). As the radicals of these amino acids are chemically different, they could not serve as phosphorylation sites. However, these positions are conserved between diatom γ-tubulin groups, hinting to their importance for some other function.

### 3.2. Diatom Tubulin Structure and Evolution

The conserved a.a. identified in the predicted sequences showed that there was common a.a. for all groups, while almost every group contained a.a. characteristic only for their group (Appendix A). A comparison of phylogenetic reconstructions (Figure 2, Figure 3 and Figure 4) and these conserved a.a. positions in diatoms showed that changes occurred in the primary structure of proteins from one phylogenetic group of tubulins to another. However, there were distinctive amino acids that were not conserved. This finding clearly confirms that each of the tubulin classes had an ancestral form in which some of the positions were formed and supported by selection. Thus, for α-tubulins, these are C75, Y87, I92, T141, S168, S172, T201, L209, V212, A232, T303, V378, L388, and A400 (Appendix A; Figure 2). β-tubulins have 12 such positions: M139, Y167, Q198, A201, A230, A232, T236, C248, C313, S323 A351, W378 (Appendix A; Figure 3), while such positions have γ-tubulins nineteen, but there are too few data on this tubulin class and changes may occur with an increase in the sample (Appendix A; Figure 4). In all tubulin classes, the groups of pennate species, which are the youngest in the evolution of diatoms (α5, β5, β6, and γ4), have the maximum difference. In β-tubulins, of the 31 positions we identified in this work for the β1 group, 12 positions in β6 are retained in the evolutionarily subsequent groups. A.a. that were most supported by selection during the diatom evolution were indicated in phylogenetic reconstructions (Figure 2, Figure 3 and Figure 4). Since some of these a.a. occurred in evolutionarily older α1 and β1, and some, upon the emergence of subsequent groups and are supported in younger ones, we assume that they may be of functional importance and, possibly, are one of the factors influencing the species-specificity of tubulins and their role in the morphogenesis of diatoms. This conclusion is also confirmed by a division of tubulins between centric (classes Coscinodiscophyceae and Mediophyceae) and pennate (class Bacillariophyceae) diatoms.

In γ-tubulins, due to the small sample of sequences, it is difficult to reliably identify conserved a.a. positions. However, even in this case, there is a noticeable difference in the composition of conserved a.a. more evolutionarily earlier and later groups. Thus, we can suppose that during the evolution of diatom tubulins, some amino acid residues were formed that are characteristic of individual groups of a certain systematic position. In the absence of experimental data, we cannot presume the function of each of them; however, we believe that maintaining conservatism in these positions may indirectly indicate their functional significance.

Based on the identified diversity of tubulin groups and the analysis of the a.a. sequences, we have assumed that tubulins of diatoms evolved independently (Figure 5). Diatoms with radial symmetry from class Coscinodiscophyceae, the earliest class, contain three α-tubulin groups (α1, α2, and α3), three β-tubulin groups (β1, β2, and β4), and a single γ2-tubulin group. The class Mediophyceae appears later, inheriting the same three Coscinodiscophyceaeα-tubulin groups (α1–α3), four β-tubulin groups (β1–β4), and three γ-tubulin groups (γ1–γ3). Our dataset includes only one species from the subclass Urneiophycidae (basal araphids), *Asterionellopsis glacialis*. This species has inherited the α3 and α4 groups of α-tubulin and the β4 group of β-tubulin and acquired the α4 group of α-tubulin during its evolution. The divergence of Urneiophycidae was followed by the formation of the class Bacillariophyceae, subclass Bacillariophycidae, in which all previously presented tubulin groups disappeared and new α5, β6, and γ4 tubulin groups surfaced. The youngest diatoms core araphids from the subclass Fragilariophycidae inherited the α4 group of α-tubulin, β4 and β5 groups of β-tubulin, and γ4 group of γ-tubulin.

The analysis performed allows us to trace changes in the structure of tubulin in diatoms. The presence in the same genomes of some species of different groups of α- and β-tubulins (*Leptocylindrus danicus*, *T. pseudonana*) confirms their independent evolution in diatoms of the class Mediophyceae. It is possible that some tubulins of certain groups as a result of duplication could acquire different properties, which subsequently led to the formation of a new tubulin group. The most surprising results regard the diatom species of the class Bacillariophyceae (Figure 5). For this class, the evolution of α-tubulins becomes dependent and excludes the presence of two groups of tubulins. It is noted that the diatom species of the class Bacillariophyceae (Figure 5) are characterized by the presence of only one group of α-, β-, or γ-tubulins. Most likely, the absence of any variations in tubulins in the Bacillariophyceae diatom genomes indicates the need for compatibility of the α- and β-tubulins. Thus, diatom tubulins could evolve both concurrently between α- and β-tubulins (Bacillariophyceae), as suggested for insect tubulins [88], and independently (Mediophyceae), which was previously shown [8].

## 4. Materials and Methods

### 4.1. Identification of Diatom Tubulin Sequences

Tubulin sequences were identified in the published genomes of *Fragilaria radians* (*Synedra acus* subsp. *radians* (Kützing) Scabitchevsky), *Phaedactylum tricornutum*, *Thalassiosira pseudonana*, *Thalassisosira oceanica*, *Pseudo*-*nitzschia multiseries*, *Pseudo*-*nitzschia multistrata*, and *Fragilariopsis cylindrus*, and in transcriptomic assemblies provided by MMETSP (Marine Microbial Eukaryotic Transcriptome Sequencing Project) [89] (Appendix A) using BLAST at the e-value cutoff of 1 × 10^−35^. Only sequences with complete, uninterrupted open reading frames (ORFs) were included for the analysis of MMETSP data [90]. We used the α- (GeneBank: NP_001257328), β- (GeneBank: NP_110400), and γ- (GeneBank: NP_001061.2) tubulin sequences from *H. sapiens* and *Arabidopsis thaliana* to query these genomes. Sequence identities were calculated on the ExPASy web server using Clustal Omega [91]. 

### 4.2. Alignment and Comparative Sequence Analysis

All identified amino acid sequences for diatom tubulins were aligned with MAFFT v7 [92]. The alignments of the identified diatom α-, β-, and γ-tubulins are available in the Appendix A, respectively). Structural conservation mapping (Figure 1B–D) was performed on the ConSurf 2016 webserver [93]. All variable positions are numbered according to the human α- (GeneBank: AAA91576.1), β- (GeneBank: BAB63321.1), and γ- (GeneBank: NP_001061) tubulin positions that they align with.

Posttranslational modification sites in predicted amino acid sequences were found using previously published information on the position and functional importance of these sites in α-, β-, and γ-tubulins of other organisms [2,12,73,74,75,76,77,78,79,80,81,87], and other references in discussion.

### 4.3. Phylogenetic Analysis

Sequence alignment for phylogenetic analysis was obtained using MAFFT v7 [92] with the parameter—maxiterate 1000, and gaps were removed with trimAL [94], at a gap threshold of 0.75. Maximum likelihood trees were built using IQ-TREE v 1.6.12 [68], with the substitution model selected using the built-in ModelFinder method [95]. Branch support was tested using SH-like aLRT, with 10,000 replicates number of bootstrap replications 10,000, minimum threshold to keep branches in the consensus tree 0.7. Tubulin isoforms from *H. sapiens* and *A. thaliana*, which were obtained from the Kyoto Encyclopedia of Genes and Genomes (KEGG) database [96], were used as an out group. The phylogenetic trees were visualized using the server i-TOL v5.7 [97]. 

## 5. Conclusions

Tubulins and other elements of the cytoskeleton are currently becoming one of the significant objects in the study of molecular and cellular mechanisms of diatom morphogenesis. This work, based on the analysis of the predicted amino acid sequences of diatom α-, β- and γ-tubulins, allowed us to identify tubulin group characteristic of certain classes and subclasses of diatoms. Groups of α1, β1 and γ1 (or γ2) tubulin centric diatoms were the first to form from the ancestral forms of these proteins. We assume that the α- and β-tubulins of diatoms of the classes Coscinodiscophyceae and Mediophyceae had a single ancestral form and subsequently evolved in parallel. The data indicate that during evolution in α-, β-, or γ-tubulins, specific post-translational modification sites and characteristic amino acid positions were maintained by selection. Along with this, each of the groups had its own amino acids, which became conserved for the particular group. The molecular evolution of α-, β-, or γ-tubulins of diatoms is comparable to the evolution of diatoms themselves and could be one of the pathways in the formation of morphological diversity.

## Figures and Tables

**Figure 1 ijms-23-00618-f001:**
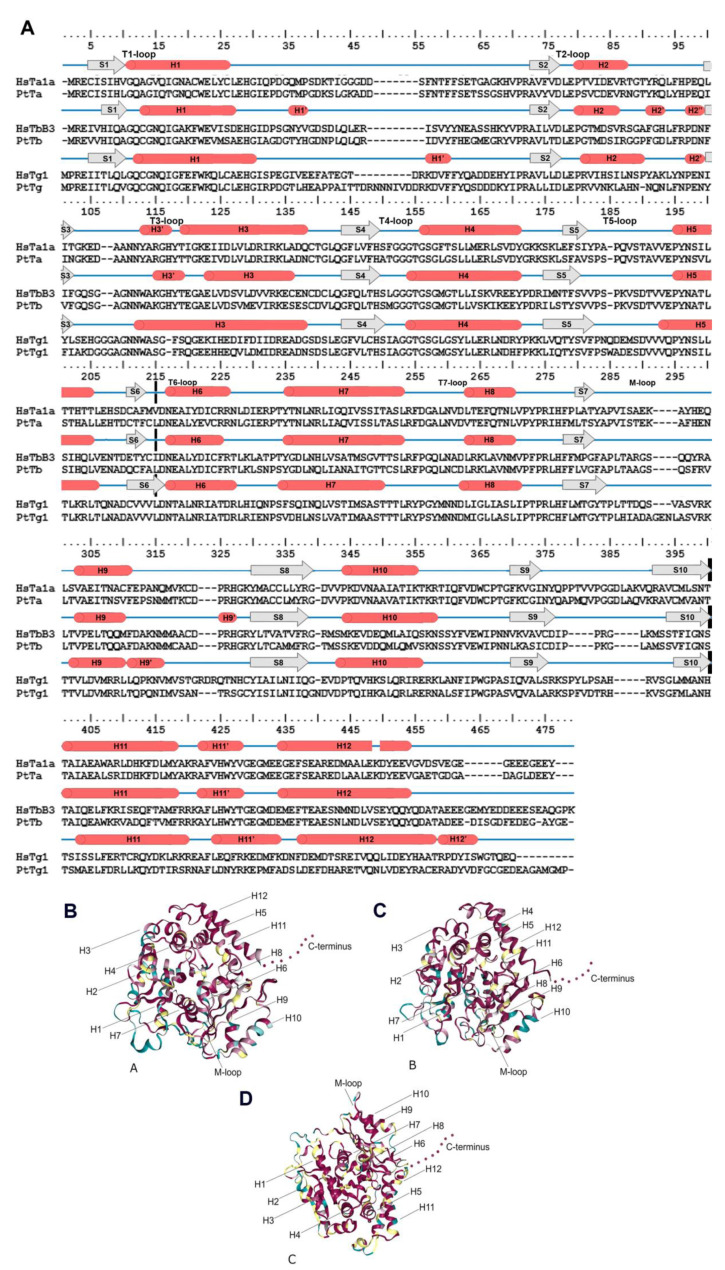
Tubulin structure. (**A**): Secondary structure for α- and b-tubulin for example *H. sapiens* (HsTa1A—α-tubulins, HsTb3—β-tubulins and HsTg1—γ-tubulin) and *P. tricornutum* (PtTa—α-tubulins, PtTb—β-tubulins and PtTg1—γ-tubulin). β-chains are marked S1–S10 and 12 α-helices—H1–H12. Cross black lines delimit major domains. (**B**–**D**): Structural conservation mapping performing on the ConSurf 2016 webserver; this conservation was assayed for all diatom tubulins. (**B**): α-tubulins; (**C**): β-tubulins; (**D**): γ-tubulins. Conserved amino acid residues are shown in shades of red, and the variable amino acids are showed in shades of blue. Residues with insufficient data are showed in shades of yellow.

**Figure 2 ijms-23-00618-f002:**
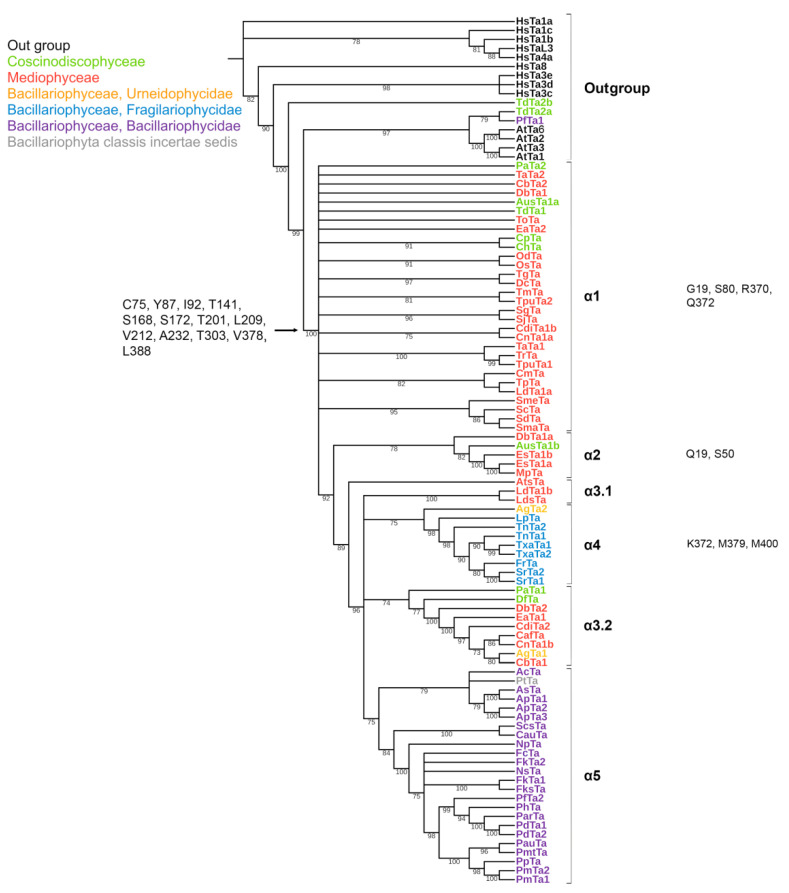
Phylogenetic tree of diatom α-tubulins. A total of 101 sequences were used to build a Maximum Likelihood tree using the LG + F + R7 substitution model. The amino acids conserved in diatoms are on the left , characteristic amino acids of certain α-tubulin groups are on the right. Large taxa are highlighted in color: green, Coscinodiscophyceae; red, Mediophyceae; blue, Bacillariophyceae, Fragilariophycidae; orange, Bacillariophyceae, Urneidophycidae; purple, Bacillariophyceae, Bacillariophycidae.

**Figure 3 ijms-23-00618-f003:**
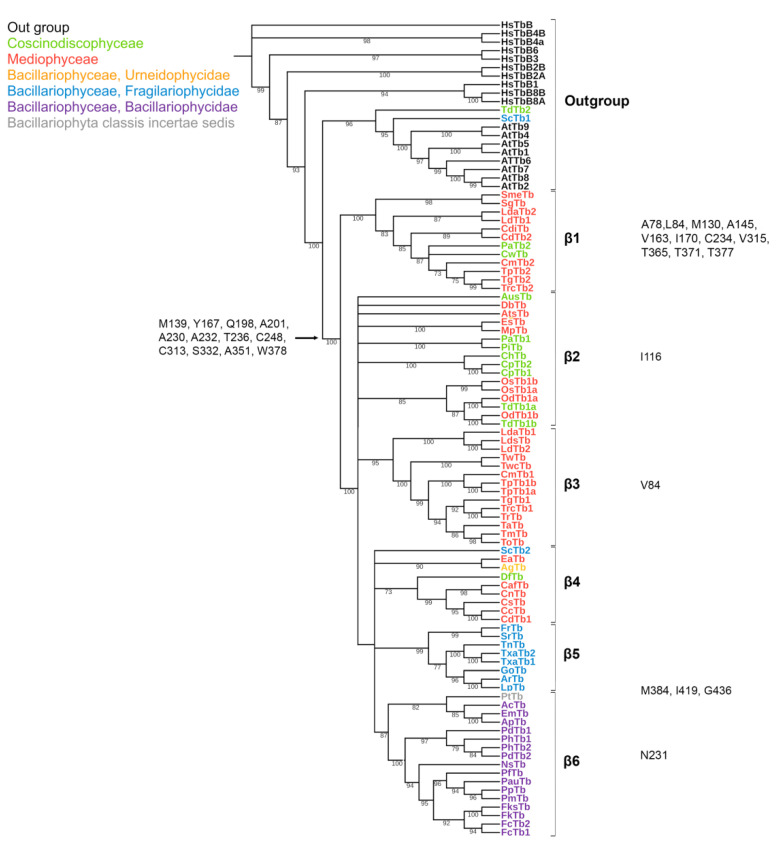
Phylogenetic tree of diatom β-tubulins. A total of 106 sequences were used to build a Maximum Likelihood tree using the LG + F + R7 substitution model. The amino acids conserved in diatoms are on the left, characteristic amino acids of certain β-tubulin groups are on the right. Large taxa are highlighted in color: green, Coscinodiscophyceae; red, Mediophyceae; blue, Bacillariophyceae, Fragilariophycidae; orange, Bacillariophyceae, Urneidophycidae; purple, Bacillariophyceae, Bacillariophycidae.

**Figure 4 ijms-23-00618-f004:**
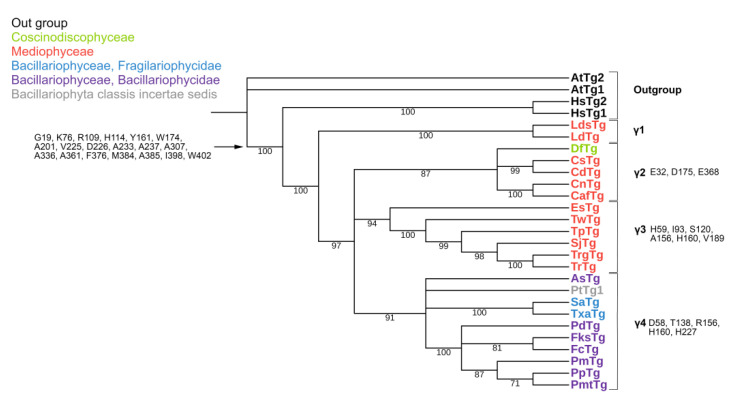
Phylogenetic tree of diatom γ-tubulins. A total of 40 sequences were used to build aMaximum Likelihood tree using the LG + F + I + G4 substitution model. The amino acids conserved in diatoms are on the left, characteristic amino acids of certain γ-tubulin groups are on the right. Large taxa are highlighted in color: green, Coscinodiscophyceae; red, Mediophyceae; blue, Bacillariophyceae, Fragilariophycidae; purple, Bacillariophyceae, Bacillariophycidae.

**Figure 5 ijms-23-00618-f005:**
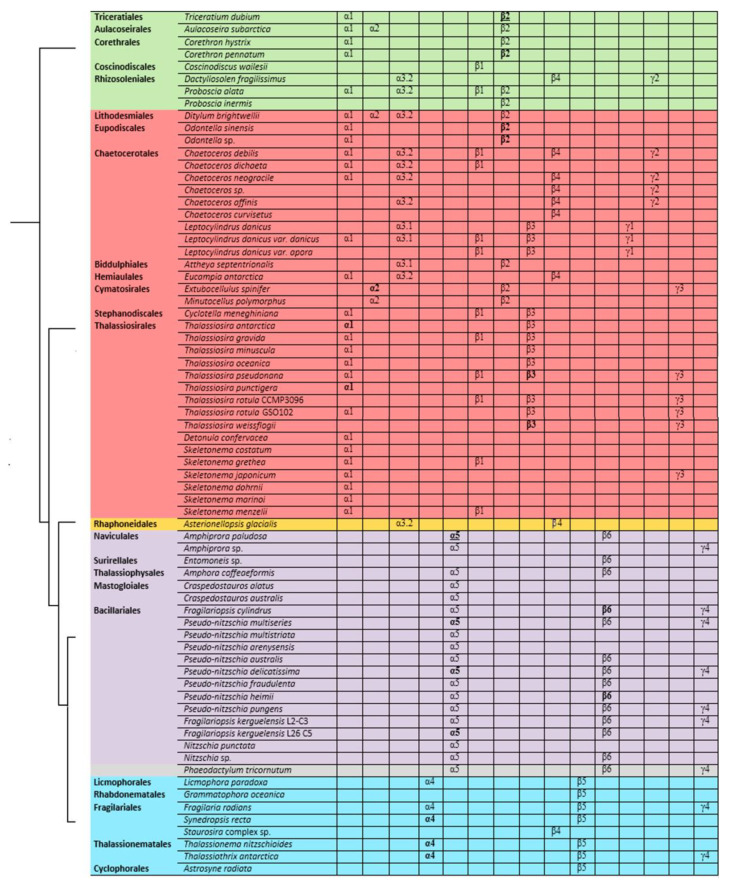
General scheme of evolution of α-, β-, and γ-tubulin in diatoms. Species names are placed according to [44]. Groups that have two copies within a given organism are highlighted in bold, and groups that have three copies are in bold and underlined.

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
