# Peer review of "Molecular Evolution of Tubulins in Diatoms"

_ijms, 2022, doi:10.3390/ijms23020618_

Round 1

Reviewer 1 Report

This is a preliminary work, to help future experiments intending to elucidate the role of microtubules in diatoms morphogenesis, in which the tubulin groups defined in this manuscript could be helpful.

Authors first extracted from genomic and transcriptomic diatom databases the sequences encoding tubulins alpha, beta and gamma. They then analyzed the sequence conservation with human and plant tubulins and found residues present in diatoms only. Phylogenetic trees of alpha, beta and gamma tubulins allowed a classification of diatoms tubulins into groups with characteristic residues, and those tubulin groups reflected well the evolution of diatoms from the oldest Coscinodiscophyceae to the more recent Bacilloriophyceae. Finally, authors analyzed the conservation in diatoms of the tubulin residues subject to post-translational modifications and found tubulin groups keeping or not those residues, suggesting distinct regulations of microtubules among diatoms. All those results enhance the interest in studying more deeply the link between microtubules and morphogenesis in diatoms groups.

The authors need to clarify some points and manuscript needs minor and major modifications before being considered for acceptance at Int. J. Mol. Sci. English should also be revised.

Major comments:

In lines 131-136: The 14 residues classified as “characteristic” of diatoms are issued from a comparison to only human and plant tubulins. I wonder to what extent those residues are characteristic of diatoms? Are they already present in the kingdom of chromists, in other plylum/phyla than diatoms? For example, one of the four alpha tubulins of Trypanosoma brucei (XP_001218940.1) shares with the Cosconodiscophycean Triceratium dubium alpha tubulin TdTa2b 9 out the 14 residues located at the diatoms’ “characteristic” positions. Nevertheless, such score does not seem to be so high with the alpha tubulins from other diatom groups. This fits with the position of TdTa2b into the outgroup in the phylogenetic tree in Figure 2.

Authors should either tune down the terms “specific” and “characteristic” or select some representative non-diatom chromist tubulins to check if their 14 alpha, 12 beta, 19 gamma tubulin residues in order to confirm (or not) that those residues are indeed specific of diatoms and not shared with chromists, humans and Arabidopsis. If those residues are present in diatoms and not in other representative of chromists, then the specificity can be stated and their importance becomes stronger than actually.

In line 247 and lines 316-317: Authors state that acetylation site is “only” retained/preserved in group α1. This should be tuned down. Indeed, even if almost all tubulins of α1 group (28/30; 93%) conserve the K40, the ratio is of  20% (1/5) for tubulins of α2 group, 33% (4/12) for tubulins of α3 group, and 21% (5/24) for tubulins of α5 group. This is not negligible. Authors could rather comment that α4 group is completely devoid of the equivalent of human K40, and that regulation of alpha tubulin by acetylation at that position is absent in those diatoms (with an expected impact on microtubule stability, as acetylated K40 is a marker of microtubule stability?). At the opposite, acetylation of alpha tubulin is probably essential in group α1 where 93% of diatoms conserve the involved Lysine, while the remaining groups present an intermediary conservation (20-33%).

In lines 322-323: Authors assume that E-to-D substitution is synonymous for polyglutamylation. It is in fact totally the opposite. Lateral glutamylation does not occur on Aspartate residues and E-to-D mutagenesis is used to generate “non-polyglutamylatable” tubulin as negative controls (see example in Wloga et al., 2008. Glutamylation on alpha-tubulin is not essential but affects the assembly and functions of a subset of microtubules in Tetrahymena thermophila. Eukaryot Cell. 7:1362-72).

In supplementary Table S7a and S7b: Authors did not analyze the correct residues for conservation of polyglutamylation and polyglycylation sites. Analysis and Tables must be redone with the correct residues.

- Tubulin branching points for polyglutamylation are listed in Fukushima et al., J. Neurochem. (2009) 109, 683–693. Authors should use the main site Glu445 of mouse alpha-tubulins, and the additional site Glu443 of mouse alpha-4 tubulin to redo the search conservation in diatoms. For beta tubulin, Glu435 and Glu438 and a putative third one (Glu441) are polyglutamylated in mouse. At least the two validated sites can be used to redo the search of conserved sites in diatoms for beta tubulins polyglutamylation.

- Tubulin branching points for polyglycylation are listed in Redeker et al., 1994. Science 266:1688-91. Authors should use the main alpha tubulin Glu445 (and putative Glu446 and Glu448) and the main beta tubulin Glu437 (and putative Glu438, Glu439 and Glu441) polyglycylated sites to redo the search conservation in diatom’s tubulins.

Minor comments:

In lines 60-61: in the microtubule field, tubulin monomers are not considered to have (+) or (-) ends. It refers in fact to the microtubule ends that exhibit structural and functional asymmetries. Structural asymmetry is because microtubule does not expose the same tubulin subunit at its ends: a β-tubulin terminates the microtubule protofilaments at the (+) end while it is an α-tubulin at the (-) end. Functional asymmetry is due to the much stronger tendency of the (+) end to polymerize and grow, in comparison to the (-) end for which this tendency is weak. The sentence must be changed to something like “Tubulin heterodimers assemble longitudinally to form protofilaments, that associate then laterally to form a sheet which progressively closes in a tube, to generate the microtubule.”

In Figure 1 (A): To be exhaustive, could HsTg1 and PtTg1be also aligned with their secondary structure above the alignment, as it is done for α- and β-tubulins? For clarity, the three alignments for α-, β- and γ-tubulins would probably be separated and not mixed as it is actually.

In Figure 1 (B-D): I guess that conservation was assayed is for all diatoms’ tubulins, as stated in the text line 124, but it should also be mentioned in the legend for clarity.

In lanes 126-127: The statement “γ-tubulins are only conserved within the helices, whereas the loops featured multiple variable sites” is not so obvious on figure 1 (B-D), as it is not possible to move the structures. The authors should generate modified “S3_Alignment_Alpha_diatoms”, “S4_Alignment_Beta_diatoms” and “S4_Alignment_Gamma_diatoms” files with the secondary structure above the alignment and an asterisk over the variable residues. It will be thus much easier for the reader to do the link between conservation and structure.

In Figure 2: To be exhaustive, the outgroup lacks a human alpha tubulin (NP_525125; TUBA3D) and an Arabidopsis alpha tubulin (NP_193232; TUA6). Those tubulins should be incorporated in the phylogenic tree and in the Supplementary Table S1.

In Figure 2: Residue S172 lacks in the specifically conserved residues listed on the left.

In Figure 3: To be exhaustive, the outgroup lacks a human beta tubulin (NP_001345618; TUB8B). This tubulin should be incorporated in the phylogenic tree and in the Supplementary Table S1.

In Supplementary Table S1: In the outgroups of human beta and gamma tubulins, is it there a reason why available accession numbers NP_821080 (HsTbB2B), NP_006079 (TUBB4B), NP_821133 (TUBB), NP_006077 (TUBB3), NP_001290453 (TUBB6) and NP_057521 (TUBG2) were replaced by a “no”?

In Supplementary Table S1: Use preferentially NP_001276052 (TUBB4A) and NP_110400 (TUBB1) instead of EAW69079 and Q9H4B7, respectively.

In Supplementary Table S1: What is the difference between bolded names versus non-bolded names? Example: Psemu1|235379 (5th lane, 2nd column) and Psemu1|4917 (6th lane, 2nd column).

In Supplementary Table S1: For outgroups of alpha, beta and gamma tubulins, also indicate the % of sequence identity between proteins issued from human or Arabidopsis paralog genes. The question behind is does diatoms present more, same, or less variability in tubulins than human and Arabidopsis.

In Supplementary Table S1 legend (line 459): “in the diatom genomic and transcriptomic data, as well as for human and Arabidopsis outgroups” in place of “in the diatom transcriptomic data”.

In Supplementary Table S2: Title does not fit and refers more to Supplementary Table S1. The criteria used for gene classification in this table need to be in the title and indicated in the legend.

In Supplementary Table S2: “out” needs to be explicated in the legend. I guess that “out” in the alpha and beta columns means that the tubulin genes/proteins fall out of their α group (α1-α5) or β group (β1-β6).

In Supplementary Tables S6a: In the human alpha tubulin AAA91576 that serves as reference for the numbering, according to the Materials and Methods, residue 232 is a Serine and not Glycine. To be changed in the table.

In Supplementary Tables S6a, S6b, S6c: Gray boxes of the actual lane 2 refer to the diatom-specific residues (for example, 14 in alpha), and you could add boxes of lighter gray for the group-specific residues (for example, 6 in alpha). What are the remaining residues indicated in the table (for example, 8 in alpha)? Do they correspond to the polymorphisms mentioned in the main text (as in line 144 for alpha tubulin)? For clarity, it needs to be stated in the legend and distinctly color-boxed.

In lines 311-313: alpha tubulin deTyrosination/Tyrosination lacks in the list of PTMs.

In Supplementary Table S7a: At 3rd lane, 2nd column: change R40 to K40 as it is a Lysine that is present in human alpha1 A/B tubulins. In the same lane, numbering is lacking for Y451.

In Supplementary Table S7a legend (and applies for S7b and S7c): Title should be simpler, such as “Conservation in diatoms of the human alpha tubulin post-translationally modified residues”.

In lines 325-326: Authors made a mix with acetylation (that occurs on Lysines) and palmitoylation (that occurs on Cysteines). I guess that the authors wanted to refer to the work of Dompierre et al. 2007, about the Brain Derived Neurotrophic Factor (BDNF) transport by motors that are more recruited on acetylated microtubules. If it is the case, “palmitoylation” must be replaced by “acetylation” and a correct reference must replace ref. [72]. If it is not the case, then the authors must develop and be more explicit.

Line 417-418: tubulins alpha AAH83344 and beta AAI47699 refers to mouse murine and not human proteins. Indicate the right reference or species used for the query.

English / Typography (examples):

  • “high” instead of “highly” (occurs often in manuscript)
  • “the last is enriched” instead of “this latter being enriched” line 144
  • “loop” instead of “loops” lane 164
  • “that involved in” instead of “that are involved in” in line 165
  • sentence line 395
  • “out group” instead of “outgroup” in text, figures and tables.
  • “incolour” instead of “in colour” in figures 2, 3, 4
  • cyrillic character in lane 165
  • some species names need to be in italic.

Author Response

We thank the Reviewers for the work on reading and reviewing our manuscript. We appreciate insightful comments and suggestions offered by the Reviewers were immensely helpful.

All changes in the text are highlighted in review mode. Clean text version is also attached.

Responses to reviewers’ comments are as follows.

Reviewer 2 Report

The article "Molecular Evolution of Tubulins in Diatoms" reads well.

Authors have performed data mining on available sources of data (genomic and transcriptomic) to gather their results.

I don't have that much issues with the article. Some parts could be improved (intro and discussion) for what concerns the language, as could be seen below.

Otherwise my only problem might be that I don't see any reference to the works of Tesson and Hildebrand (https://journals.plos.org/plosone/article?id=10.1371/journal.pone.0014300). It has been one of the main work on diatoms microtubules for years.

I listed a few suggestions below:

11 highly conserved?

12 higly variable?

253 H. sapiens should be in italics. Same thing at 315.

325: 'Lysine palmitation site' it is palmitoylation here, no?

329: 'turned out to be conserved both diatoms and
human' I guess some 'among' is missing here.

332: won't it be better to change 'showing' for 'suggesting'?

335-336 'but it
is known four phosphorylation sites (S80, S131, T288, and S361 [81]) for other organisms.' Might it be better to write 'four phosphorylation sites (S80, S131, T288, and S361 [81]) are known among other organisms'?

382: 'Coscinodiscophycaea' I guess Coscinodiscophyceae instead

395: 'The analysis performed, it can be observed the history of the diatom tubulin de-
velopment.' Sentence sounds odd.

400: 'The most surprising that diatom species of
the class Bacillariophyceae' odd too

I wish the authors good luck for the rest of the submission process.

Author Response

(The authors gave the same response as above.)

Round 2

Reviewer 1 Report

The author made substantial corrections of the manuscript. Nevertheless, there are still two major points that remains and need to be corrected. Lines numbering for the comments is issued from file “ijms-1481722-peer-review-v2.pdf”:

Major points:

1 / Lines 275-276: Concerning alpha tubulin polyglutamylation, the reference 65 (Fukushima et al., 2009) highlights residues E445 as the major site and E443 as a minor site. Authors only considered the minor site E443 and not the main site E445, despite its conservation in group in Mediophyceae + Coscinodiscophycidae and a weak conservation in Fragilariophycidae + Urneidophycidae. This should be commented in the text, as it is in favor of a selective conservation of polyglytamylation sites. The study still lacks the analysis for the conservation of polyglycylation sites E446 + E448E for alpha tubulins, and 437 + E439 of beta tubulins.

Consequently, in Table S7a,

  • Please remove the column “C-terminal EE” as it is totally uninformative given the absence of numbering.
  • Please rename column for E443 with “Polyglutamylation”.
  • Please add two columns (E446 and E448), named “Polyglycylation”, for which authors have to fill the homology with diatoms.

Consequently, in Table S7b:

  • Please rename the columns E435 “Polyglutamylation”.
  • Please rename the columns E438 “Polyglutamylation/Polyglycylation”.
  • Please add two columns (E437 and E439), named “Polyglycylation”, for which authors have to fill the homology with diatoms

Consequently, in lines 371-372: sentences must be replaced by a comment about the conservation results for polyglytamylation and polyglycylation sites, based on the above comments on E445 polyglutamylation and on the requested results for polyglycylation of alpha tubulin E446 and E448, and of beta tubulin E437 and E439.

2 /Lines 285-286 and Table S7a: Authors state that the palmitoylation site is K376 in human and that it is not conserved in diatoms. The 376th residue of human alpha tubulin (NP_001257328) is a Cys and not a Lysine.

In the paper Ozols and Caron (1997) referenced by the authors, it is stated that the main palmitoylation site of porcine alpha tubulin is C376 in the peptide VQRAVCM (see the Table in the Fig 5A of the cited paper). This Cysteine and the palmitoylated peptide are conserved in humans and diatoms (VQRAVCM in human and VKRAVCM in PtTa, under the s10 sheet in figure 1A). Thus, the main palmitoylation site of Ozols and Caron (1997) is conserved in human (at C376) and at least in P. tricornutum.

This is a positive result enhancing the conservation of PTMs between diatoms/plants/human. Consequently, the authors must re-analyze the human C376 conservation with the correct corresponding residue in diatoms, and re-fill the palmitoylation column of table S7a. Lines 285-289 and 362-365: sentences must consequently be replaced by a comment about the conservation results for C362 i(the one of VQRAVCM in PtTa) in diatoms, based on the conservation results.

I suppose that the authors made a mix in residues numbering, especially if they used their figure 1A, as there is only one numbering line for 3 alignments (alpha, beta and gamma tubulins) and as the alignments also introduce gaps that complicate the residue numbering reading. To facilitate the understanding for the readers, I strongly encourage the authors to separate the three alignments (Fig 1A for alpha, Fig 1B with beta, Fig 1C with gamma, and Fig 1D-F for structures). They should also put:

  • Above each alignment, the numbering ruler for the human sequence (as it serves of reference, taking care of the gaps that diatom tubulins create and the structure elements (helix, sheet, loop)
  • At the right of each sequence, a number indicating the position of the last residue of the row

Minor points:

In manuscript and in figures: “outgroup” should be in place of “out-group”

Lines 134, 312: in these cases, “highly” was correctly used in fact. I should have been more precise in my comment at the first round of review, by indicating for which lines it applied.

Line 266: “2.4. Analysis of Post-Translational Modification Sites” should be in bold and aligned to the left.

Line 276: Authors should indicate out of how many species polyglutamylation site E443 is conserved, and cite again ref 65.

Lines 299 and 301: Do authors mean that aspartate can substitute glutamate? In that case, they should put “to aspartate” instead of “to glutamate”.

Lines 279-280: Modified text is bolded. “tyrosine (Y) residue and polyglutamylation (addition of E to γ-carboxy group of E side chains and chain elongation by further addition of E residues) or polyglutamylation (addition of G to γ-carboxy group of E side chains and chain elongation by further addition of G residues) are conserved for all groups (Supplemental Table S7a).”

Line 360: «glutamylation» should be in place of «glutamination»

Lines 521, 526, 531: “helixes” should be in place of “spirals”

Figure 2: change “AtTa4” to “AtTa6”.

Figure 2: “S172” is still missing in the list of residues specific to diatoms, on the left of the tree.

In Table S1: in the line for HsTg2, change accession number “NP_0B5” to “NP_057521”.

Author Response

We are grateful to the reviewer for his work and patience. The changes made the manuscript more understandable and informative. All changes in the text are highlighted in review mode. Clean text version is also attached.

Responses to reviewers’ comments are as follows.

Round 3

Reviewer 1 Report

I thank the authors for their patience and for the modifications provided. Once the three minor modifications below will be done in the text, the manuscript will be suitable for publication in Int. J. Mol. Sci.

Minor changes

Lines 268-269: remove “se-quences”

Lines 285-288:  change the sentence “Sites of detyrosination in C-terminal tyrosine (Y) residue and glutamylation or polyglutamylation (addition of E to γ-carboxy group of E side chains and chain elongation by further addition of E residues) are conserved for all groups (Supplemental Table S7a)” by the sentence ” Sites of detyrosination in C-terminal tyrosine (Y) residue and polyglutamylation (addition of E to γ-carboxy group of E side chains and chain elongation by further addition of E residues) or polyglutamylation (addition of G to γ-carboxy group of E side chains and chain elongation by further addition of G residues) are conserved for all groups (Supplemental Table S7a).”

Line 292: change “K376” by “C376”

Author Response

Dear editors and reviewer,

My colleagues and I are grateful for your work on our manuscript. We accepted all the proposed edits and made them into the text. A clean version of the text is attached.

Sincerely,

Yekaterina Bedoshvili.